# Older Adults and Clutter: Age Differences in Clutter Impact, Psychological Home, and Subjective Well-Being

**DOI:** 10.3390/bs12050132

**Published:** 2022-05-03

**Authors:** Helena L. Swanson, Joseph R. Ferrari

**Affiliations:** Department of Psychology, College of Science and Health, DePaul University, Chicago, IL 60614, USA; hswanso2@depaul.edu

**Keywords:** psychological home, clutter, subjective well-being, aging

## Abstract

Previous research found mixed results for clutter’s impact on individuals’ sense of home and subjective well-being in a variety of samples. In this retrospective cross-sectional study, archival data were utilized to examine the relationship between clutter, psychological home, and subjective well-being across two age categories, specifically older adults aged ≥65 (*n* = 225), and younger adults aged ≤64 (*n* = 225). Three moderation analyses used age categories as a moderator exploring the relationship between (a) clutter predicting psychological home, (b) psychological home predicting subjective well-being, and (c) clutter predicting subjective well-being. Results found that age categories significantly moderated the relationship between clutter and psychological home but did not moderate the other variable relationships.

## 1. Introduction

Understanding what impacts people’s sense of home and subjective well-being is important for researchers to identify, as we seek to examine individual and community health. Previous research found that one phenomenon that impacts an individual’s sense of home and subjective well-being tends to be high levels of clutter, “an overabundance of possessions that create chaotic and disorderly living spaces” (Roster et al.; para. 2) [1]. Roster and colleagues [1] found support for a theoretical model of psychological home and latent variables, including clutter and subjective well-being. In addition, previous literature found a significant relationship between clutter, a sense of psychological home, and subjective well-being, with a variety of different samples, such as women of color, persons who are indecisive, and high anxiety [1,2,3,4,5,6]. However, to date, no previous research has compared older adults’ (≥65 years old) and younger adults’ (≤64 years old) experiences with clutter, sense of psychological home, and subjective well-being.

### 1.1. Psychological Home, Subjective Well-Being, and Clutter

Psychological home is defined as “a sense of belonging in which self-identity is tied to a particular place” [7] (p. 11). Sigmon and colleagues [7] detail three functions of psychological home: a place of refuge that provides security, safety, protection, and privacy from the external world; the development of object attachments and place attachments in the physical environment to support self-identity; and greater psychological subjective well-being. A key aspect of psychological home is the ability to change one’s physical environment to support self-identity [8]. When the physical environment hosts possessions that are problematic to the owner (e.g., clutter), these possessions might impact a person’s perspective of home [1,2,4]. For example, Crum and Ferrari [3] found that clutter partially mediates the relationship between psychological home and subjective well-being.

Subjective well-being is composed of three components, including life satisfaction, positive affect, and negative affect [9]. For decades, researchers and the media have noted a relationship between happiness and age, citing that older adults are generally happier and report greater subjective well-being than younger adults ([10,11,12,13,14,15,16,17], as cited in [18]). However, not all research findings are consistent with the claim that older adults are happier or have greater subjective well-being than younger adults. For example, Bowling [19] found that individuals, regardless of age, report similar definitions of subjective well-being by saying that the main drivers of subjective well-being are self-rated health, mental health symptoms, long-standing illness, and social support. Yang and colleague [18] found that age and happiness have a J-shaped relationship, where younger adults were somewhat happier than middle-aged adults, with happiness declining throughout middle age, and happiness reaching a nadir at 40 years old; after 40 years old, people regain their happiness, with an upward trajectory for the latter half of life. Furthermore, Yang and colleague [18] found that the relationship between age and happiness was weakly moderated by health status and strongly moderated by income.

Most previous clutter research has utilized life satisfaction as the objective definition for subjective well-being when testing clutter’s relationship to subjective well-being [1,2,5,6]. In addition to subjective well-being, the research on clutter found that excessive items impact on many other aspects of individual’s lives; including employee burnout/tension and satisfaction with one’s job [20,21], indecision (decisional procrastination) and behavioral procrastination [22], need for cognition [5], and psychological home [1,2,5,6]. Results from previous literature testing the relationships between clutter, psychological home, and subjective well-being concluded that the more clutter someone has, the less sense of psychological home and subjective well-being they report [1,2,5,6].

Previous research that found a relationship between clutter, sense of psychological home, and subjective well-being had samples that included emerging and younger adults [2,4,23], women of color [3], employees [21,22], college students [4,5], and middle-aged adult samples [1,4,6]. When comparing three different samples, college students (*M* age = 21), younger adults (age range: 18–44, *M* age = 31), and middle-aged adults (age range: 21–84, *M* age = 54), Ferrari and Roster [4] found that clutter issues were negatively related with satisfaction with life, but only for the middle-aged sample. While Ferrari and Roster (2018) conducted analyses that included some representation of older adults (≥65 years old), they did not examine age cross-sectionally to analyze whether older adults differed from younger adults within their “middle-age sample”. Similarly, Prohaska and colleagues [5] found in a sample of undergraduate college students where 66% of the sample was 24 years old or younger that there was no relationship between age and clutter asserting that clutter accumulation does not increase with age in their young adult sample. Furthermore, Girts [23] did not find a significant relationship between clutter accumulation tendencies and life satisfaction with a sample of emerging adults.

However, to date, no previous research has compared older adults’ (≥65 years old) and younger adults’ (≤64 years old) experiences with clutter, sense of psychological home, and subjective well-being. Taken together, these previous research findings provide a basis for questioning whether the established relationship between clutter, psychological home, and subjective well-being may be experienced differently based on age, specifically in the later part of life (e.g., older adults, 65+).

### 1.2. Life Course Perspectives

Utilizing a life course perspective has become an increasingly popular direction for many interdisciplinary researchers over the last 30–40 years [23]. Researchers that frequently use life course perspectives have sought to gain consensus on the topic ([24,25,26], as cited in [23]). One consensus about life course perspectives is that “changes in human lives (as changes in personal characteristics and transitions between states) are considered over a long stretch of lifetime, such as from childhood to old age…” ([23]; p. 414). An emerging topic in life course research is health and well-being across the life course, and its subsequent changes. As detailed above, clutter impacts on individuals’ well-being; therefore, this provides grounding to the importance of studying clutter and its impacts among various age groups to assess potential life course changes.

The present study is influenced by life course perspective theories, and Roster and colleagues’ [1] theoretical model for psychological home, clutter, and subjective well-being for age differences. As such, the present study will further examine Roster and colleagues’ [1] theoretical model for age differences, as noted in Figure 1. More specifically, this project will examine whether there are age differences in experiences with the relationships between clutter, sense of psychological home, and subjective well-being for older adults (65+ years old), and a random sample of younger adults (≤64 years old). Traditionally in gerontology literature, 65 years old is used as the cutoff between younger adults and older adults; as such, this study utilized 65 years old as the cutoff age to divide age categories. Results from this study will bridge multiple professional fields, including social psychology, gerontology, professional organizing, clinical psychology, and sociology, through the addition of crucial context to research on clutter and its related variables, by highlighting aging differences.

### 1.3. Hypotheses

The present study has three hypotheses following Roster and colleagues’ [1] theoretical model.

**Hypothesis** **1.***Age categories (older adults: ≥65 years old; younger adults: ≤64 years old) moderate the relationship between clutter and psychological home*.

**Hypothesis** **2.***Age categories (older adults: ≥65 years old; younger adults: ≤64 years old) moderate the relationship between psychological home and subjective well-being*.

**Hypothesis** **3.**
*Age categories (older adults: ≥65 years old; younger adults: ≤64 years old) moderate the relationship between clutter and subjective well-being.*


## 2. Method

This study is a retrospective cross-sectional study utilizing archival data that were previously collected online in 2015. The study was approved in 2013 by the University of New Mexico’s Institutional Review Board under the study ID 13-212. Data were collected as part of a larger study examining home environments and clutter. Participants (*N* = 1394) were recruited with the help of a partnering organization, the Institute for Challenging Disorganization [27] (ICD). ICD posted the survey link and an invitation on their website homepage for survey recruitment. Participants had to be 19 years or older to participate, and the age range was 21 to 84 years old (*M* = 54 years old, *SD* = 11.28). All participants provided informed consent before participating in the study.

### 2.1. Participants

To test for age differences, two samples were extracted from the data: the total present sample for older adults (65+, *n* = 225, age range: 65–84, *M* age = 69.84, *SD* = 4.65) and a random sample of younger adults (≤64 years old, *n* = 225, age range = 23–64, *M* age = 52.04, *SD* = 9.11). Examining the age distributions between the two age categories revealed that approximately 30% of the younger adult category was younger than 50 years old, and approximately 52% of the older adult age group was aged 68 or younger. The total extracted study sample (*N* = 450) was a majority of female (94.0%), White, non-Hispanic participants (93.0%), married (56.0%), who lived in a detached single-family house (68.9%), and had a total household income of more than $50,000 (64.7%). Table 1 displays the demographic breakdowns for each of the subsamples.

### 2.2. Psychometric Scales

Psychological home (PSYH) was measured using the eight-item Psychological Home Scale developed by Sigmon and colleagues [7]. The psychological home scale measures participants’ relationship with their physical home environment, including the benefits of having a sense of psychological home. Responses for the Psychological Home Scale ranged from 1 (strongly disagree) to 7 (strongly agree). Sample items include “I have grown attached to many of the places I have lived” and “I take pride in the place where I live”. Table 2 includes the scale mean, standard deviation, and reliability.

To measure how clutter negatively impacts participant lives, the 18-item Clutter Quality of Life Scale (CQLS), developed by ICD [27], was utilized. The CQLS was developed to understand the extent to which individuals perceive their clutter as negatively impacting them. CQLS responses ranged from 1 (strongly disagree) to 7 (strongly agree). Sample items include “The clutter in my home upsets me” and “I feel overwhelmed by the clutter in my home”. Table 2 includes the scale mean, standard deviation, and reliability.

Diener and colleagues’ [28] five-item Satisfaction with Life Scale (SWLS) measured subjective well-being. SWLS is intended to measure participants’ evaluation of their life in terms of satisfaction or feelings of fulfillment, by asking questions about life experiences and emotions associated with experiences. Responses ranged from 1 (strongly disagree) to 7 (strongly agree). Sample items are “In most ways my life is close to my ideal” and “I am satisfied with my life”. Table 2 includes the scale mean, standard deviation, and reliability.

Lastly, the 13-item Marlow–Crowne Social Desirability Scale [29] (MCSD) was used as a self-report “control variable,” to ascertain whether respondents answered in an overly socially appropriate way. The scale is designed to be a forced-choice response with true–false responses for each item, to ascertain whether participants are responding in an overly socially desirable way (i.e., not being truthful). Sample items include, “I’m always willing to admit when I make a mistake” and “I have never been irked when people expressed ideas very different than mine”. Table 2 includes the scale mean, standard deviation, and reliability.

## 3. Results

### 3.1. Analysis Plan and Preliminary Results

All necessary assumptions for testing the study hypotheses were met. All project analyses used jamovi [30,31]. Zero-order Pearson’s correlations indicated that MCSD was significantly related to two of the three study variables, CQLS (*r* = 0.147, *p* = 0.002) and SWLS (*r* = −0.244, *p* < 0.001), indicating that participants may have responded in a socially desirable way for those two measures. As such, partial correlations [32] were conducted for the three study variables (clutter, psychological home, subjective well-being, and social desirability) controlling for social desirability, and are presented in Table 2.

Preliminary *t*-test analyses examined age category differences for the four target self-report scale measures. An independent *t*-test between age category differences for CQLS revealed that younger adults (*M* = 4.42, *SD* = 1.66) experience significantly more negative CQLS ratings compared to older adults (*M* = 4.11, *SD* = 1.58), *t*(448) = −1.99, *p* = 0.048. Another independent *t*-test evaluating age category differences for SWLS indicated that older adults (*M* = 4.51, *SD* = 1.53) experience significantly higher positive life satisfaction compared to younger adults (*M* = 4.15, *SD* = 1.50), *t*(448) = 2.58, *p* = 0.010. For PSYH, a *t*-test revealed that older adults (*M* = 5.80, *SD* = 0.90) report a significantly greater sense of psychological home compared to younger adults (*M* = 5.50, *SD* = 1.02), *t*(448) = 3.27, *p* = 0.001. Lastly, an independent *t*-test concluded that younger adults (*M* = 6.82, *SD* = 2.78) were more likely to give socially desirable survey responses than older adults (*M* = 5.43, *SD* = 2.79), *t*(431) = −5.18, *p* < 0.001.

### 3.2. Primary Analyses

To evaluate the first hypothesis, a moderated linear regression analysis was conducted using clutter as the predictor variable, psychological home as the outcome variable, and age categories (older adults: ≥65 years old; younger adults: ≤64 years old) as the moderator. The analysis concluded that there was a significant negative interaction between clutter and age, impacting psychological home, β = −0.11, *SE* = 0.05, *p* = 0.032. The model significance indicated that younger adults experience the negative relationship between clutter and psychological home more intensely than older adults.

To evaluate the second hypothesis, a moderated linear regression analysis was conducted using psychological home as the predictor variable, subjective well-being as the outcome variable, and age categories as the moderator. There was no significant interaction observed, β = −0.23, *SE* = 0.14, *p* = 0.106. However, psychological home had a significant positive direct effect on subjective well-being, β = 0.45, *SE* = 0.07, *p* < 0.000. Age categories did not have a direct effect on subjective well-being, β = −0.23, *SE* = 0.14, *p* = 0.087.

To evaluate the third hypothesis, a moderated linear regression analysis was conducted, using clutter as the predictor variable, subjective well-being as the outcome variable, and age categories as the moderator. There was no significant interaction observed, β = −0.05, *SE* = 0.08, *p* = 0.497. However, clutter had a significant negative direct effect on subjective well-being, β = −0.46, *SE* = 0.04, *p* < 0.000. Age categories did not have a direct effect on subjective well-being, β = −0.23, *SE* = 0.12, *p* = 0.065. Figure 1 displays the full theoretical model with the present study coefficients. The results for the moderation analyses are also presented in Table 3.

## 4. Discussion

This study contributes to the previous literature on clutter, psychological home, and subjective well-being; see Roster et al. [1]. The results provide an insight into how age is an important variable when considering the relationship between clutter, psychological home, and subjective well-being. Older adults reported significantly greater subjective well-being and psychological home, as well as significantly less social desirability, compared to younger adults. The present study analyses support the Roster et al. [1] theoretical model, as all direct effects in Figure 1 follow the same pattern as the original model. More specifically, clutter negatively predicted psychological home, psychological home positively predicted subjective well-being, and clutter negatively predicted subjective well-being.

We conducted three moderation analyses consistent with Roster and colleagues’ [1] theoretical model, to further investigate potential age category differences. The study results indicate that age category differences were only present for the relationship between clutter quality of life and psychological home (Hypothesis 1). These results highlight that clutter negatively impacts psychological home for younger adults more than older adults, despite both groups experiencing a negative impact on the psychological home as a result of clutter. The significant interaction does not necessarily mean that older adults have more or less clutter items than younger adults, but simply that older adults’ clutter impacts their sense of home differently compared to younger adults. This may be because as we age, our items increasingly become a form of self-extension, to the point where we do not see clutter items as being problematic, but as part of ourselves and our home. The other two moderation analyses revealed that psychological home (positively) and clutter (negatively) impact upon subjective well-being similarly across age categories.

Theories in sociology are frequently concerned with life course transitions [23,33], one of the transitions being the transition to older adulthood and its associated changes [34]. It is possible that through the life transition from younger adulthood to older adulthood, the drivers of psychological home may change, making clutter less impactful for psychological home for older adults.

### Limitations and Future Research

The current study is not without its limitations. The present study’s sample had an overwhelming majority of White, non-Hispanic females. However, previous research by Prohaska and colleagues [5] tested the Roster et al. [1] theoretical model with a non-White, urban sample, and found results supporting the model. Nevertheless, it is possible that an intersection of race and age could change the impact of model variables. Future research should seek to obtain a diverse study sample (including race, gender, and age) to further test the theoretical model. Additionally, the results indicate that younger adults were more likely to respond in a socially desirable way compared to older adults, and so the Hypothesis I significant interaction may be due to younger adults in this sample responding dishonestly. Another study limitation is the narrowly distributed representation of ages in the two age categories. Future research should seek to obtain a sample that equally represents participants of all ages, to ensure a proper distribution of ages amongst younger and older adult categories. Furthermore, future research should explore individuals’ drivers of psychological home, and whether the drivers change with age. Future research would also benefit from reviewing these phenomena through an interdisciplinary lens, including fields such as public health, community psychology, and gerontology.

## 5. Conclusions

Having a home that we self-identify with, that makes us feel safe and secure, and that provides solitude from the external world is crucial for our subjective well-being. Understanding how age differences may impact subjective well-being is important when seeking to support different individuals’ sense of psychological home. A breadth of literature links clutter with a decreased sense of psychological home. This study is the first to add to that literature by adding age context, specifically differences between older and younger adults, showing that younger adults experience the negative impact of clutter for psychological home more intensely than older adults; however, both groups’ psychological home is negatively impacted by their clutter. The authors suggest that aging should be considered as an important factor in research involving clutter and psychological home.

## Figures and Tables

**Figure 1 behavsci-12-00132-f001:**
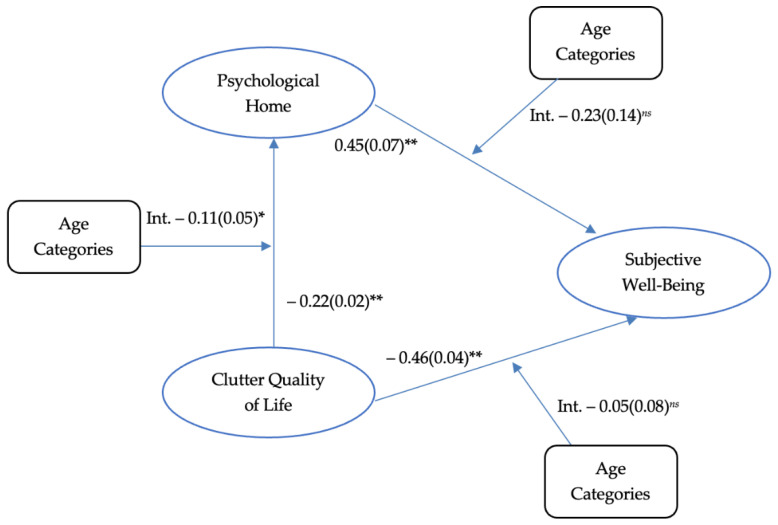
Theoretical Model of Psychological Home with Present Study Coefficients. *Note:* Direct effect estimates are on the inside of the figure. Interaction effects are indicated with “Int.” before statistics. Statistics presented: β(*SE*). *ns* = *p* > 0.05, * *p* < 0.05, ** *p* < 0.001.

**Table 1 behavsci-12-00132-t001:** Demographic Breakdown for Subsample.

Demographic Category		Older Adults (*n* = 225)		Younger Adults (*n* = 225)	
		*n*	%	*n*	%
Gender					
	Female	206	92.0%	216	96.0%
	Male	18	8.0%	9	4.0%
Race					
	White, non-Hispanic	208	94.5%	203	91.4%
	Black, non-Hispanic	5	2.3%	5	2.3%
	American Indian or Alaska Native	2	0.9%	1	0.5%
	Asian/Pacific Islander	1	0.5%	2	0.9%
	Hispanic/Latino	3	1.4%	8	3.6%
Relationship Status					
	Married	114	51.4%	135	60.5%
	Divorced/Separated	46	20.7%	30	13.5%
	Single	22	9.9%	34	15.2%
	Widowed	35	15.8%	12	5.4%
	Partnered/Cohabitating	5	2.3%	12	5.4%
Housing					
	Own dwelling	180	81.1%	173	77.2%
	Rented dwelling	38	17.1%	44	19.6%
Residence Type					
	Detached single-family house	159	71.3%	165	73.7%
	Townhouse/Condominium	25	11.2%	25	11.2%
	Apartment	28	12.6%	24	10.7%
	Duplex	3	1.3%	5	2.2%
	Manufactured/Mobile Home	3	1.3%	5	2.2%
	Other	5	2.2%	0	0.0%
Cohabitation by Age					
	6 or younger	3	2.0%	20	13.6%
	7–12 years old	4	2.7%	30	19.4%
	13–17 years old	2	1.4%	29	19.6%
	18–25 years old	5	3.5%	37	25.5%
	26–35 years old	11	7.5%	20	14.1%
	36–45 years old	9	6.0%	45	28.7%
	46–55 years old	3	2.1%	87	54.7%
	56–65 years old	48	30.3%	114	67.1%
	66–75 years old	160	78.4%	10	7.8%
	76+ years old	45	31.0%	6	4.9%
Household Income					
	Less than $20,000	16	9.2%	10	5.3%
	$20,000–$34,999	30	17.3%	13	7.0%
	$35,000–$49,999	23	13.3%	35	18.7%
	$50,000–$74,999	38	22.0%	39	20.9%
	$75,000–$99,999	31	17.9%	32	17.1%
	$100,000 or more	35	20.2%	58	31.0%

*Note:* Not all participants responded to all demographic questions. For the Cohabitation demographic category, *n* values represent how many participants indicated having at least 1 person of each age group in their home.

**Table 2 behavsci-12-00132-t002:** Zero-order and Partial Correlates (controlling for social desirability) between Self-Reported Scales.

	*M* (*SD*)	1	2	3	4
CQLS	4.26 (1.63)	[0.97]	−0.490 **	−0.375 **	-
2. SWLS	4.33 (1.52)	−0.498 **	[0.91]	0.292 **	-
3. PSYH	5.65 (0.98)	−0.376 **	0.294 **	[0.86]	-
4. MCSD	6.12 (2.86)	0.147 **	−0.244 **	−0.029	[0.69]

*N* = 450 ** *p* < 0.001. *Note*: CQLS: Clutter Quality of Life Scale, SWLS: Satisfaction with Life Scale, PSYH: Psychological Home. Values along the diagonal are coefficient alpha representing scale reliability with the present total sample. Value above the diagonal is partial correlates, and value below the diagonal is zero-order correlates.

**Table 3 behavsci-12-00132-t003:** Moderation Results.

	β	*SE*	*Z*	*p*
Model 1:				
CQLS	−0.22	0.023	−8.83	<0.001 ***
Age Categories	−0.23	0.09	−2.67	0.007 *
CQLS * Age Categories	−0.11	0.05	−2.26	0.024 *
Model 2:				
PSYH	0.45	0.07	6.45	<0.001 ***
Age Categories	−0.23	0.14	−1.71	0.087
PSYH * Age Categories	−0.23	0.14	−1.62	0.106
Model 3:				
CQLS	−0.46	−0.04	−12.03	<0.001 ***
Age Categories	−0.23	0.12	1.85	0.065
CQLS * Age Categories	−0.05	0.08	−0.68	0.497

*Note:* Model 1 corresponds with Hypothesis 1 (IV: CQLS, DV: PSYH, Moderator: Age Categories); Model 2 correspond with Hypothesis 2 (IV: PSYH, DV: SWLS, Moderator: Age Categories); Model 3: corresponds with Hypothesis 3 (IV: CQLS, DV: SWLS, Moderator: Age Categories). * *p* < 0.05; *** *p* < 0.001.

## Data Availability

Data sharing not applicable.

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
