# Peer review of "Older Adults and Clutter: Age Differences in Clutter Impact, Psychological Home, and Subjective Well-Being"

_behavsci, 2022, doi:10.3390/bs12050132_

Round 1

Reviewer 1 Report

Hi to the authors

Thank you for the opportunity to review this paper.

This paper covers a topic that continues to baffle researchers: the relationship between clutter, psychological home, and subjective well-being across age groups, with findings across studies showing inconsistencies. I felt that the paper is well written and convincingly argued, that the design, hypotheses and analysis are mostly clearly stated and appropriate, and that the findings and discussion are coherent. I believe that the paper would be of interest to the journal's readership, however in my view it requires revision around the following points to strengthen its contribution.

  1. The introduction is reasonably comprehensive and offers the reader the background to this study at a sufficient breadth and depth. However, few points require clarification, more detail or revision in my view.
    1. The introduction confuses between two terms: well-being and subjective well-being. Unfortunately, this is a common source of confusion, and a significant number of published papers make a similar error. As new publications draw on such papers, this perpetuates the confusion. The authors used the satisfaction with life scale, which is an aspect of happiness - commonly referred to in the literature as hedonic or subjective wellbeing. While happiness can be seen as an aspect of well-being, it does not represent the key components of well-being - often referred to as eudaimonic wellbeing (which reflects items such as purpose and meaning in life, relationship, autonomy, growth, engagement, accomplishment and the like). Since the authors correctly used in much of the paper the term “subjective well-being”, this source of confusion can be easily rectified, by using the term subjective well-being / happiness / satisfaction with life consistently (instead of well-being) in the title, in the introduction, and in the discussion, and also by making the same distinction when referring to cited papers.
    2. When discussing subjective well-being, please cite earlier research that reports on the association between age and subjective well-being / happiness / satisfaction with life.
    3. While the authors cite a few papers who found that age is statistically correlated with clutter and psychological home (and other related constructs) there is little theoretical grounding as to how, why and in what way age matters, and what are the key age groups that are theoretically distinct. I’d recommend offering some detail on one theoretical model (or more) that can help frame and situate the paper and its findings theoretically. One such model is briefly described in the discussion – it may be useful to describe it in a bit more detail in the introduction.
    4. I’m a bit baffled by this sentence which is attempting to highlight the contribution of the current paper: “However, to date, no previous research compared older adults’ (≥ 59 65 years old) and younger adults’ (≤ 64 years old) experiences with clutter, sense of psychological home, and subjective well-being”. Compared to the research cited earlier as well as later on, it looks like other studies did refer to age, as well as covered all age groups. Hence I’d suggest wording this sentence in a way that makes the contribution of this paper clearer and more distinct.
    5. My key concern regarding the paper is related to the above point: it’s contribution and scientific value is not currently worded in a convincing way that would make it distinct enough to merit a publication.
    6. I’m unsure if this is the journal’s instructions, but if not, I’d suggest moving the hypotheses to the end of the introduction section.
  2. Method section: the method section is adequate and most parts are reasonably detailed. The following points require clarification or more detail in my view.
    1. In the participants section it would be useful to offer the mean, standard deviation, distribution and range of ages included in each of the clustered age groups. Without these details it is difficult to assess how the two groups differ in terms of age? Given the similarities shown later on in table 1 in their socio-economic background and residence type, there is a sense that the age gap between the two groups may be quite small? The details could be added to table 1.
    2. Another point, which would be highly useful to mention if the authors have access to this type of data, is whether all those included (in both groups) own their home or rent, and whether they live with parents (for the young adults) or in a shared accommodation (both young and older adults) or with family (for older adults). Intuitively it seems that such variables would directly impinge on people’s a sense of home and the capacity to handle clutter.
    3. I’m unsure if this is the journal’s requirement, but if not, I’d suggest moving table 2 to the results section.
    4. I’d suggest adding age to table 2 (using the full range of age rather than the groupings).
    5. It seems that a data analysis section is missing? Please add.
    6. Was a normality test conducted? Since parametric analyses are used, please add the results of a normality test.
    7. Please clarify which type of correlation was used?
    8. Was a reliability analysis conducted? It states consistently that table 2 shows the reliability of all scales – but it seems to be missing.
  3. Results section: the section is coherent and well structured. The following points require clarification or more detail in my view.
    1. It would be useful to see the results of the regressions in a table, rather than in written form, though the chart is useful and clear.
    2. In section 3.2 I’m not sure why age was used in the regression as a categorical moderating variable rather than using its full range? My thinking is that it may yield a higher B value if used to its full range? Unfortunately, being aware of today’s competitive publishing arena, despite being a well written paper, the editors may feel that a B of -.011 (p=0.05) is fairly weak moderation power to report on, particularly as it comes across as the key finding and novelty of the paper.
    3. Another question is whether a regression was run to predict satisfaction with life by clutter and psychological home within each group separately? Given that there were significant differences between the groups on all variables, another way to test the hypotheses is to run the regressions in this way to assess how the groups differ.
  4. Discussion: the section is generally well written and ties reasonably well the findings of this paper with earlier work. The following points require more careful wording in my view.
    1. In my view some sentences in the discussion take the findings (which are not very strong) a bit too far in terms of interpretation. I’d recommend a more nuanced wording.
    2. The following sentence is unclear: “Of the three moderation analyses conducted to test Roster and colleagues [1] theoretical model for age category differences only Hypothesis I had a significant interaction. Results from Hypothesis II and III revealed that psychological home (positively) and clutter (negatively) impacting well-being were experienced similarly across age groups.” I’d suggest rewording.
    3. More explanation of the findings (or lack of) would be helpful in my view to help strengthen the theoretical contribution of the paper.
    4. As noted clarifying earlier the mean, standard deviation, distribution and range of ages included in each of the clustered age groups would have been useful to explain the findings.
    5. Theories concerned with life course transitions are mentioned in the discussion briefly. To make fuller use of these, I’d suggest referring to them in in more detail in the introduction.

I hope that you find these notes useful and wish you all the best.

Reviewer 2 Report

I have reviewed this paper and it looks okay, particularly how younger adults experience the negative impact of clutter on the psychological home more intensely than older adults.

However, the study methodology is a bit confusing and it would be good for the authors to spell it out, especially on line 84. This study sounds like a retrospective cross-sectional study and if it is, let it reflect on line 84. It also needs to be included in the abstract so that it is clear in the summary.

Round 2

Reviewer 1 Report

Thank you for addressing my notes so promptly. I reviewed the paper and happy with  the revisions. Wishing you lots of luck with the publication!